# Development of Multiplex Polymerase Chain Reaction (PCR)-Based MSA Assay for Bladder Cancer Detection

**DOI:** 10.3390/ijms241713651

**Published:** 2023-09-04

**Authors:** Thomas Reynolds, Maxie Gordon, Gabriela Vanessa Flores Monar, David Moon, Chulso Moon

**Affiliations:** 1NEXT Bio-Research Services, LLC, 11601 Ironbridge Road, Suite 101, Chester, VA 23831, USA; 2HJM Cancer Research Foundation Corporation, 10606 Candlewick Road, Lutherville, MD 21093, USA; 3BCD Innovations USA, 10606 Candlewick Road, Lutherville, MD 21093, USA; 4Department of Otolaryngology-Head and Neck Surgery, The Johns Hopkins Medical Institution, Cancer Research Building II, 5M3, 1550 Orleans Street, Baltimore, MD 21205, USA

**Keywords:** bladder cancer, loss of heterozygosity, microsatellite instability, triplet MSA assay

## Abstract

Several studies have shown that microsatellite changes can be profiled in the urine to detect bladder cancer. Microsatellite analysis (MSA) of bladder cancer detection requires a comprehensive analysis of up to 15–20 markers based on amplifying and interpreting many individual MSA markers, which can be technically challenging. To develop fast, efficient, standardized, and less costly MSA to detect bladder cancer, we developed three multiplex polymerase chain reaction (PCR) based MSA assays, all of which were analyzed by a genetic analyzer. First, we selected 16 MSA markers based on nine publications. We developed MSA assays based on triplet or three-tube-based multiplex PCR (Triplet MSA assay) using samples from Johns Hopkins University (JHU Sample, first set of samples). In the second set of samples (samples from six cancer patients and fourteen healthy individuals), our Triplet Assay with 15 MSA markers correctly predicted all 6/6 cancer samples to be cancerous and 14/14 healthy samples to be healthy. Although we could improve our report with more clinical information from patient samples and an increased number of cancer patients, our overall results suggest that our Triplet MSA Assay combined with a genetic analyzer is a potentially time- and cost-effective genetic assay for bladder cancer detection and has potential use as a dependable assay in patient care.

## 1. Introduction

Microsatellite instability (MSI) is a molecular tumor phenotype resulting from genomic hypermutability and manifested by genome-wide variations in the length of microsatellite sequences. As a familiar part of colon cancer syndrome, MSI is common in individuals with Lynch syndrome [1,2,3]. However, since its discovery, MSI has been a generalized phenomenon in a wide spectrum of sporadic cancers [4,5,6,7,8]. The underlying mechanisms of sporadic cancers may be based on epigenetic mechanisms, namely MLH1 methylation. The accumulation of frameshift mutations in tumor-associated genes is the most detrimental outcome of MSI, either in inherited or sporadic tumors. These mutations can activate crucial pathways promoting human carcinogenesis. Determining whether a tumor exhibits microsatellite instability (MSI) is useful in identifying patients with hereditary non-polyposis colorectal cancer and sporadic gastrointestinal cancers with defective DNA mismatch repair (MMR). Diagnosing MSI in cancer is currently achieved by examining polymerase chain reaction (PCR) products from a few (typically 5–7) informative microsatellite markers (MSI–PCR) [9,10].

Bladder cancer (referred to as transitional bladder carcinoma) is a major cause of morbidity and mortality in the Western world [11]. Based on our understanding of its etiology, bladder cancer is an ideal candidate for screening due to molecular characteristics associated with disease progression and management guidelines [12,13]. Due to the lack of disease-specific symptoms, diagnosis of and follow-up for bladder cancer remains a challenge for the medical community, not only urologists but also primary care physicians [13,14,15]. Cystoscopy, the gold standard for detecting bladder cancer, is invasive and costly, sometimes carrying unwanted complications. Urine cytology is of limited value due to its low sensitivity, particularly to low-grade tumors. New qualitative and quantitative molecular tests have been designed to identify cellular and subcellular alterations exclusively associated with bladder cancer [14,15,16]. Therefore, over the last two decades, several new “molecular assays” for diagnosing urothelial cancer have been developed. Overall, combining several molecular assays with traditional screening methods has demonstrated promising results [16,17,18,19,20]. Based on histopathology and clinical presentations, two different types of noninvasive bladder cancer have been classified, including the frequently recurring papillary tumor (Ta) and the more aggressive carcinoma in situ (CIS). While either type can progress into invasive tumors (T1–T4), the chance that low-grade Ta tumors progress to invasive disease is much less likely than high-grade Ta tumors and CIS. It is commonly accepted that noninvasive tumors (that is, Ta and CIS) can be grouped into ‘superficial bladder cancers’ and are distinguishable from invasive tumors, which invade the basement membrane [17,18,19].

Loss of heterozygosity (LOH) is typically identified by comparing DNA isolated from tumors to germ-line DNA, such as that isolated from blood. It is important to recognize that Microsatellite instability and LOH are two different genomic instability types, one is microsatellite and the other is chromosomal, and their etiology is different. For example, in colorectal tumors, tumors with microsatellite instability have completely different paths and outcomes compared to chromosomal instability tumors. Overall, LOH can be detected by gene scan, which differs from microsatellite instability analysis [1,2,3]. Abnormalities involving p16 (chromosome 9p21) and p53 (17p13) are associated with superficial transitional cell carcinoma (TCC). These loci are the two most common regions of LOH in bladder cancer [20,21,22,23,24,25,26,27]. LOH at 9p is known to have prognostic value in non-muscle-invasive bladder cancer. Other loci implicated in bladder cancer progression have also been characterized, involving chromosomes 18q, 4p, 16, 20, and 21 [28,29,30,31,32,33,34,35]. LOH is typically identified by comparing DNA isolated from tumors to germ-line DNA, such as that isolated from blood. Short tandem repeat (STR) regions, also known as microsatellite regions within chromosomes, are unstable in cancerous cells and are consequently deleted, causing a loss of heterozygosity (LOH) in the tumor sample. Microsatellite instability analysis (MSA) targets, such as tandem repeats in genomic DNA, are one method of evaluating loss of heterozygosity (LOH) in tumor cell transformation [24,25,26]. These biomarkers, originally developed at Johns Hopkins University, represent a panel of fifteen short tandem repeat (STR) or microsatellite regions that detect deletions in DNA isolated from the urine sediment of bladder cancer patients before cystoscopic evidence of a tumor [34,35,36]. Several studies showed that these microsatellite changes can be profiled in urine to detect bladder cancer cells [37,38,39,40,41,42,43,44]. MSA combines 15– 20 markers from regions with a high percentage of LOH. This method has proven to be overly sensitive for low- and high-grade lesions with sensitivities of 67, 86, and 93% for recurrent G1, G2, and G3 lesions, respectively, and a specificity of 88% [37,38,39,40,41,42,43,44]. Moreover, MSA could predict recurrence before cystoscopic detection in all studies with extended follow-up [44]. MSA is a highly accurate, automated assay, but careful validation from “multi-center studies” and “exclusion of patients with persistent leukocyturia” is crucial for it to become clinically applicable. The discovery, validation, and translation of biomarkers for the early detection of cancer is the primary focus of the Early Detection Research Network (EDRN), an initiative of the National Cancer Institute (NCI). A prospective study sponsored by the EDRN was undertaken to determine the efficacy of a novel set of MSA markers for the early detection of bladder cancer [45,46]. Academia and industry partners collaborated to perform this work and data analysis. An analysis of samples from 500 individual participants in the study (300 cancer patients and 200 benign controls), indicated >80% sensitivity and specificity in cancer patients compared to healthy controls, exclusive of confounding variables and sample errors [45].

Microsatellite analysis for detecting cancer has several significant technical challenges, specifically in allele calling and interpretation. Lab-to-lab variability due to instrumentation and personnel causes differences in assay performance. Additionally, both stochastic effects and variations in peak height effects between DNA derived from urine sediment and blood samples create more variation, contributing to interpretation differences. These differences were especially evident in our samples, from which MSA results produced results slightly above or below the cut-off ratios established for LOH. Hence, qualifying the STR assay presents significant challenges. Determining the right parameters for LOH detection in potential tumor cells isolated from urine sediment is one of the key hurdles in establishing dependable MSA tests [34,35,36,37,38,39,40,41,42,43,44].

We developed a triple multiplex PCR-based MSA assay to develop a standardized and less costly MSA for detecting bladder cancer. In this report, we present two key data sets. Firstly, we selected 16 different markers for our assay development. Secondly, we present our final and viable MSA assay based on a triplet multiplex PCR reaction (Triplet MSA assay). We also briefly discuss MSA use in early bladder cancer detection and its role as a surveillance tool for recurrent bladder cancer.

## 2. Results

### 2.1. Selection of 16 Markers

Out of the nine publications listed in Table 1, we selected 15 different MSA primer sets for 16 markers (Table 2). These primer sets have showed dependable high LOH in bladder cancer samples with an exceptionally low number of LOH among samples from healthy individuals. For over 350 patients, the combined data provided over 90% sensitivity and 100% specificity. These 16 different markers were characterized by separate independent PCR-based MSA analyses to analyze desirable cutoff values.

### 2.2. Development of Triplet Multiplex PCR

We originally designed triplet (three-tube) multiplex PCR reaction and singlet (one-tube) singleplex PCR primer sets and compositions for individual primers in triplet (three-tube) multiplex PCR (Table 3). The primer D13S304 (AAAG)n 6-FAM GGCTGCATGAGCCCTAAGTA TGGGTGACACAGTGAGACTCTA) was not included in the final triplet MSA assay because it caused inconsistent PCR amplification. We included MP2 marker MBP in the multiplex, which generates two distinct amplified markers MBP and MBPA, totaling 15 amplified markers from 14 primer sets. Table 4 and Table 5 describe the cut-off values. We repeated this final triplet assay multiple times on various samples, including 25 cancer samples and 15 healthy control samples (JHU samples), and reproducibility was consistent. In a non-blind fashion, we tested 20 QC samples from the EDRN study with our newly established assay. As shown in Figure 1, out of these QC samples, 6/20 cancer samples were determined to be cancer based on an overall evaluation of 15 tested markers, and 14/20 samples from healthy individuals were determined to be non-cancerous.

## 3. Discussion

Considering aspects of clinical presentation in the expected course of disease progression, bladder cancer has potential value as a screening target, and there is common consensus in screening high-risk populations [13,14]. There are two key aspects that will affect the importance of bladder cancer screenings in the upcoming decades. Firstly, the persistently high prevalence of smoking is expected to be one of the key hazards to long-term urothelial carcinogenic effects for several generations. Secondly, it is highly unlikely for bladder cancer to metastasize before it becomes invasive [19,20,21], which provides a valuable opportunity for early bladder cancer detection between tumor origination and invasion. Noninvasive cancer management is associated with fewer morbidities and is more effective than managing invasive tumors [13]. In the early stages of tumor development, more aggressive treatment modalities including cystectomy, systemic chemotherapy, or chemoradiation therapy are not required.

Many reports from several independent groups previously confirmed the superior sensitivity of MSA (75–96%) compared to cytology (13–50%) in various clinical settings [35,36,37,38,39,40,41,42,43,44,47,48,49,51,52,53,54,55,56,57,58,59,60,61]. Based on an analysis of 377 cancer samples in reports from 1997 to 2001, the sensitivity was 90% and the specificity was 100% [34,35,36,42,44,45,48,49,51]. Unlike conventional cytology, microsatellite analysis (MSA) can detect low-grade and low-stage diseases as accurately as high-grade and high-stage diseases [35,36,37,38,39,40,41,42,43,44,47,48,49,51,52,53,54,55,56,57,58,59,60,61,62]. Frigerio et al. [37] found that the combined use of cytology and LOH analysis had a higher sensitivity than either test alone for identifying primary tumors and could detect almost all recurrent diseases in voided urine. Similarly, van Rhijn et al. [43] demonstrated three important findings after studying 47 patients with confirmed superficial bladder TCC (37 pTa, 10 pT1) at initial diagnosis and proposed potential clinical applications of MSA. MSA correctly identified 94% (44/47) of primary tumors and 92% (12/13) of tumor recurrences. MSA also predicted the chance of recurrence at 75% (9/12) for molecular detection 1–9 months before cystoscopy evidence of recurrent disease.

In summary, several important studies demonstrated that urine microsatellite analysis could detect superficial bladder tumors and be a potentially reliable test for detecting and predicting tumor recurrence. Moreover, detection rates can be improved in combination with urine cytology. In this sense, it is important to note the meta-analysis performed by van Rhijn et al. [58], who performed an extensive literature review using 18 markers and concluded that MSA, Immunocyte, NMP22, CYFRA21-1, Lewis X, and FISH are among the most promising surveillance markers [49,58,59,60,61,63,64,65,66,67,68,69,70]. Nevertheless, there is not enough clinical evidence to warrant the substitution of the cystoscopy follow-up scheme for any of the currently available urine marker tests. Similarly, current data are not consistent with the sole use of molecular tests in patients at high risk of developing bladder cancer. However, several studies have shown molecular tests’ value in improving the diagnostic accuracy of bladder cancer for high-risk groups and predicting recurrence. Improvements in diagnostic accuracy are only consistent when used in conjunction with cytology and cystoscopy.

We learned several important lessons from this study, most of which other studies also recognized. Several points enhancing the sensitivity and specificity of MSA warrant further discussion. Firstly, by establishing a robust genetic marker profile and determining individual threshold values for LOH/allelic imbalance in each of the analyzed microsatellites, we can maximize the sensitivity of tumor DNA detection without compromising its specificity. In this sense, we used rigorous statistical analysis based on reliable studies (Table 1). Secondly, dependable methods are necessary to avoid erroneous LOH-judgements due to PCR artifacts, which others have also described [41,42,43,44]. We initially tried two-tube-based PCR or doublet multiplex PCR, which seemed successful initially but demonstrated a lack of dependable marker amplification after repeated tests. Therefore, we decided not to pursue this assay. Thirdly, performing MSA on a genetic analyzer must always be a standard method of MSA practice as it has two major advantages. Sample processing is largely automated, and results are provided as a numerical data readout, independent of the inter-observer variability associated with the complex interpretation of morphological features. Moreover, determining LOH ratios on this platform provides reproducible and reliable results even in situations where cell conservation is suboptimal for cytological evaluation and/or FISH. In our experience, using a standard genetic analyzer such as the ABI 3100 machine provides significant advantages since MSA is unaffected by changes in PCR conditions or the amounts of genomic DNA applied as long as it meets the minimum requirements. For example, when we repeated multiple rounds of our triplex PCR, most of the markers resulted in consistent findings. The amount of genomic DNA to be analyzed must be sufficient. We propose using at least 20 to 30 ng of urine genomic DNA to be on the safe side of DNA quantity, 20 ng for 10 markers, and 30 ng for 15 markers. We used DNA amounts based on this guideline. Sometimes DNA from the urine samples of healthy individuals is <30 ng and the MSA results from both two- and three-tube PCR become inconsistent or non-reproducible. However, our reports lack some important information. For JHU and EDRN samples, we were not provided with clinicopathological information on the patient samples used for the study. This paper could benefit from more detailed information about age, sex, smoking history, clinical stage of the disease, and whether the tumors are muscle-invasive or non-invasive. Furthermore, we did not test the MSI status of the cancer samples by MSA or examine the expression of MMR proteins (hMLH1, hPMS2, hMSH2, and hMSH6) to further validate our data due to the lack of proper samples. Similarly, we did not receive information about the disease progression history of patients who tested positive for cancer by Triple MSA. We feel that our report could have provided more clarity on this newly developed assay with proper clinical information about individual cancer patients in addition to healthy controls and an increased number of samples from cancer patients (we used 25 JHU samples and 6 EDRN samples). When we initiated this study, we felt that a small number of samples with limited clinical information was sufficient for initial assay development. The report we present reflects the contract laboratory responsible for performing the MSA assay and includes MSA assay results using qualification-blinded samples to assure the assay’s accuracy before its use in the clinical trial validation study discussed above. We wish to publish these results as part of our efforts to report findings from the EDRN validation clinical trial in a follow-up paper, including a statistical analysis of each marker for sensitivity, specificity, and accuracy and refining the assay to include only the most informative markers. 

In conclusion, we presented data for developing a “Triplet MSA Assay combined covering 15 markers with a genetic analyzer” and propose its use as a potentially time and cost-effective genetic assay for bladder cancer detection. We also discussed prior data regarding MSA-based bladder cancer detection. From 20 QC samples, our triple MSA correctly predicted all 6/6 cancer samples to be cancerous and 14/14 samples from healthy individuals as non-cancerous (Table 1). This result is consistent with other reports discussed above in terms of diagnostic accuracy. We hope that our assay can potentially improve bladder cancer management.

## 4. Material and Method

### 4.1. Matched Blood and Urine Genomic DNA Samples

Blood and urine specimens for the qualification study were obtained from Dr. David Sidransky’s lab at Johns Hopkins University (JHU sample). These included 15 biopsy-proven superficial bladder cancer patients, 10 healthy controls (eligible study participants) in the assay development for MSA. These samples were previously used to determine proper cut-off values for a series of 10 MSA markers. The parent trial and secondary analysis reported in this manuscript were approved by the Institutional Review Board of Johns Hopkins University. Written informed consent was obtained from all participants and/or their legal guardians. All research was performed in accordance with relevant guidelines/regulations from Johns Hopkins University. Research involving human participants was performed in accordance with the Declaration of Helsinki. 

### 4.2. EDRN QC Samples

We designed this prospective study to determine the efficacy of MSA markers for detecting bladder cancer and bladder cancer recurrence using a newly developed clinical MSA assay [45,46]. As a part of the study, a quality control CLIA/College of American Pathology (CAP) accredited laboratory (QA Lab) and the University of Maryland Baltimore Biomarker Reference Laboratory (UMB-BRL) performed quality control analysis on 10% of the samples from the testing laboratory (QC Samples). The Institutional Review Board from EDRN approved the parent trial and secondary analysis reported in this manuscript. Written informed consent was obtained from all participants and/or their legal guardians. All research was performed in accordance with relevant guidelines/regulations from EDRN. Research involving human participants was performed in accordance with the Declaration of Helsinki. EDRN identified the sample status as cancerous or non-cancerous after the results were evaluated.

### 4.3. DNA Extraction and Quantification

Matched blood and urine samples remained unidentified during the study. The testing lab purified genomic DNA from the buffy coat of blood collected in ACD collection tubes using the QIAGEN QIAMP Blood Mini kit (Germantown, MD, USA) according to the manufacturer’s instructions. Genomic DNA from urine sediment was purified using the QIAGEN QIAMP Viral RNA Mini kit. The DNA was quantified against a standard curve of human DNA using TaqMan β-actin Detection Reagents from Applied Biosystems (Norwalk, CT, USA). We used 30 mL of urine samples and 5 mL of blood in this study.

### 4.4. STR Targets and PCR Primers

The panel of STR targets for the MSA assay was originally developed at Johns Hopkins University as a radioactive PCR assay [34]. The testing lab converted the radioactive PCR assay to a high-throughput capillary electrophoresis assay for fluorescent PCR products. The resulting assay comprised two or three multiplex reactions from which primers were selected (Table 1) and each PCR product’s primer sequence is outlined in Table 2. We obtained 5′ fluorescent primers from Applied Biosystems and 3′ primers from Integrated DNA technologies. Following round three of the qualification studies, the singleplex reaction was dropped due to poor performance in the assay. The primers were initially mixed at an equimolar concentration (2 pm for each primer). We adjusted the primer concentrations after refining the assay. Based on an assay developed for the EDRN study, five sets of multiplex PCR were designed [45,46].

### 4.5. Multiplex PCR: Triplet (Three Tube) Multiplex PCR

We ran five sets of optimization reactions (qualification) to design a triplet (three-tube) multiplex PCR reaction using primers in Table 4. For the first three rounds of qualification, the STR regions were amplified using AmpliTaq Gold DNA Polymerase (Applied Biosystems, Foster City, CA, USA). The PCR conditions were 4 mM MgCl2/0.2 mM dCTP/0.2 mM dGTP/0.4 mM dUTP/0.5 mM dATP/2 units AmpliTaq Gold. MP1 and MP2 were amplified using 4 ng of total genomic DNA, respectively, for both blood and urine DNA. MP3 and the D13S804 singleplex reaction were amplified using 6 ng of blood or urine DNA in a 25 µL final volume. PCR cycling conditions were 95 °C for 11 min/32 cycles of 94 °C for 1 min, 55 °C for 1 min, 72 °C for 1 min/60 °C for 45 min/4 °C hold. For rounds four and five of the qualification, we modified PCR amplification conditions to use 1× FastStart Taqman Probe Master (Roche, Basel, Switzerland). The DNA concentrations and PCR cycling conditions remained the same.

### 4.6. Capillary Electrophoresis

We performed capillary electrophoresis on 1 µL of PCR product using the 3100 Genetic analyzer from Applied Biosystems (Foster City, CA, USA). We ran matched blood and urine samples in the same injection to prevent difference in run conditions between matched specimens. Following electrophoresis, we analyzed the data using Gene mapper v3.1 (Applied Biosystems, Foster City, CA, USA) and the peak sizes and peak heights were exported as a tab delimited text file.

### 4.7. Calculation of Loss of Heterozygosity 

We imported the data exported from the 3100 analyzer into Microsoft Excel for analysis. In order for the data to be acceptable, the positive control DNA (HL-60 genomic DNA (ATCC)) had to pass strict peak height and size criteria before the sample data could be analyzed (Table 5). If the data met these criteria, the ratio of heterozygous alleles in blood and urine could be determined by the following formula: Ratio = ((urine 1 allele 1 peak height/urine 1 allele 2 peak height)/(blood 1 allele 1 peak height/blood 1 allele 2 peak height)). This ratio was then compared to cut-off values previously determined by calculating ratios of 50, which matched healthy blood and urine specimens.

### 4.8. Data Analysis

We performed data analysis using GeneMapperID. “Positive” results indicated a variant or missing allele at the loci indicated. Target loci are known cancer markers. Loss of heterozygosity (LOH) at the loci indicates when one of the two alleles is missing in the urine sample from a heterozygous locus in the buccal sample. LOH can only be detected at heterozygous loci; hence, homozygous loci are uninformative. The overall MSA genotype is calculated as Negative, LOH-Low (1–2 markers), LOH-Medium (3–4 markers), and LOH-High (>4 markers). Tumors typically acquire more LOH markers at advanced growth stages. Unless stated otherwise in the Validation Protocol, we applied the following acceptance criteria to determine the suitability of each assay.

### 4.9. ABI 3100 DNA Analyzer Acceptance Criteria

To be deemed Negative (no LOH detected), a heterozygous locus must have RFU peak heights between 200 and 100,000 RFUs with a peak height ratio between the cut-off values determined for each STR marker (Table 2). To be deemed LOH-Positive, the samples must have at least one heterozygous locus with RFU peak heights between 200 and 100,000 RFUs with a peak height ratio above or below the cut-off values determined for each STR marker (see Table 2). To be deemed non-informative, the sample must be homozygous at the locus. Homozygous loci must have only one peak, with sample RFU peak heights falling between 200 RFUs and 100,000 RFUs. A sample will be deemed un-evaluable if it does not meet the minimum or maximum RFU for peak height or has no signal at all. A minimum of eight loci must be analyzed and meet the Negative, LOH, or non-informative call in order to pass the sample acceptance criteria. Failed samples are only repeated once. Samples that fail a second time do not meet the sample acceptance criteria and are reported as Quantity Not Sufficient (QNS).

## Figures and Tables

**Figure 1 ijms-24-13651-f001:**
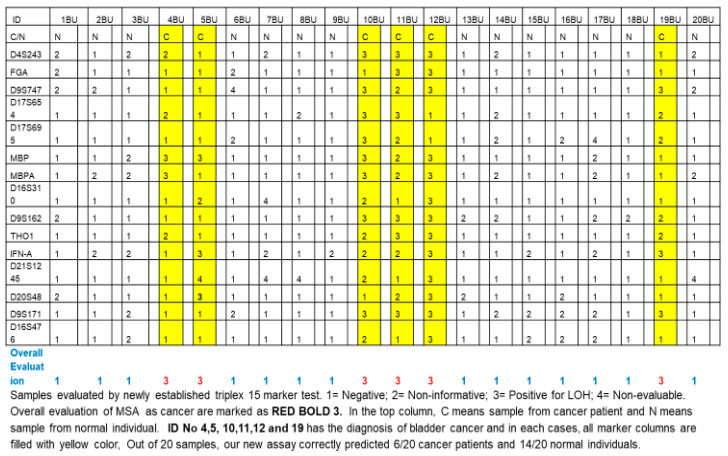
Results of validation study using triplet (three-tube) multiplex PCR.

**Table 1 ijms-24-13651-t001:** List of prior publications leading to the selection of 16 markers.

Study	No. of Cancers Detected by MSA	Sensitivity (%)	Healthy Controls with Neg MSA Result	Specificity (%)
Mao et al. (1996) ♦ Science 271:659–662 [34]	19/20	95	5 out of 5	100
Steiner et al. (1997) ♦ Nat. Med. 3:621–624 [42]	10/11	91	10 out of 10	100
Linn et al. (1997) ♦ Int. J. Cancer 74:625–629 [47]	13/15	87	N/A	N/A
Schneider et al. (2000) ♦ Cancer Res. 60:4617–4622 [48]	87/103	84	N/A	N/A
Sourvinos et al. (2001) ♦ J. Urol. 165:249–252 [49]	26/28	93	10 out of 10	100
Zhang et al. (2001) ♦ Cancer Lett. 172:55–58 [50]	73/81	90	19/19	100
Seripa et al. (2001) ♦ Int. J. Cancer 95:364–369 [36]	33/34	97	11 out of 11	100
Zhang et al. (2001) ♦ J. Natl. Cancer Inst. 93:45–50 [51]	22/23	96	17/17	100
Amira et al. (2002) ♦ Int. J. Cancer 101:293–297 [44]	44/47	94	N/A	N/A
Overall	327/362	90%	72/72	100%

MSA, Microsatellite instability analysis. N/A: not applicable.

**Table 2 ijms-24-13651-t002:** List of microsatellite instability (MSI) forward and reverse primers with their DNA sequence. FGA denotes the gene creating protein fibrinogen A, MBP for myelin basic protein, TH01 for tyrosine hydroxylase 1, and IFNA1 for interferon alpha 1.

Locus	
D13S802	Primer F: TGACACAGTGAGACTCTATCTCAAAAA
(AAAG)n	Primer R: CTTCAGACTGGCTTAGACTGAGG
D4S243	Primer F: TCAGTCTCTCTTTCTCCTTGCA
(ATAG)n	Primer R: TAGGAGCCTGTGGTCCTGTT
FGA	Primer F: GACATCTTAACTGGCATTCATGG
(TTTC)n	Primer R: CTTCTCAGATCCTCTGACACTCG
D9S747	Primer F: GCCATTATTGACTCTGGAAAAGAC
(GATA)n	Primer R: CAGGCTCTCAAAATATGAACAAAAT
D17S654	Primer F: ACCTAGGCCATGTTCACAGC
(CA)n	Primer R: GAGCAGAATGAGAGGCCAAG
D9S162	Primer F: GCAACCATTTATGTGGTTAGGG
(CA)n	Primer R: TCCCACAACAAATCTCCTCAC
D17S695	Primer F: CTGGGCAACAAGAGCAAAAT
(AAAG)n	Primer R: TTTGTTGTTGTTCATTGACTTCAGTC
MBP	Primer F: GGACCTCGTGAATTACAATCACT
(ATGG)n	Primer R: ATCCATTTACCTACCTGTTCATCC
D21S1245	Primer F: CCAGAAAATGACACATGAAGGA
(AAAG)n	Primer R: TTGTTGAGGATTTTTGCATCA
D16S310	Primer F: GGGCAACAAGGAGAGACTCT
(ATAG)n	Primer R: AAAAAAGGACCTGCCTTTATCC
D20S48	Primer F: ATGGTCTCCAGTCCCATCTG
(GT)n	Primer R: TTGACCTGGATGAGCATGTG
THO1	Primer F: AGGCTCTAGCAGCAGCTCAT
(TCAT)n	Primer R: TGTACACAGGGCTTCCGAGT
D9S171	Primer F: TCTGTCTGCTGCCTCCTACA
(CA)n	Primer R: GATCCTATTTTTCTTGGGGCTA
D16S476	Primer F: GGCAACAAGAGCAAAACTCC
(AAAG)n	Primer R: GGTGCTCTCTGCCCTATCTG
IFN-A	Primer F: TGCGCGTTAAGTTAATTGGTT
(GT)n	Primer R: GTAAGGTGGAAACCCCCACT

**Table 3 ijms-24-13651-t003:** Different primer compositions used for triplet (three-tube) multiplex PCR.

Multiplex	Target	Chromosome	STR	Forward Primer Sequence (5′-3′)	Reverse Primer Sequence (5′-3′)
MP1	D4S243	4	(ATAG)n	6-FAM-TCAGTCTCTCTTTCTCCTTGCA	TAGGAGCCTGTGGTCCTGTT
	FGA	4	(TTTC)n	VIC-GACATCTTAACTGGCATTCATGG	CTTCTCAGATCCTCTGACACTCG
	D9S747	9	(GATA)n	VIC-GCCATTATTGACTCTGGAAAAGAC	CAGGCTCTCAAAATATGAACAAAAT
	D17S654	17	(CA)n	NED-ACCTAGGCCATGTTCACAGC	GAGCAGAATGAGAGGCCAAG
	D17S695	16	(AAAG)n	PET-CTGGGCAACAAGAGCAAAAT	TTTGTTGTTGTTCATTGACTTCAGTC
MP2	D9S162	9	(CA)n	NED-GCAACCATTTATGTGGTTAGGG	TCCCACAACAAATCTCCTCAC
	MBP	18	(ATGG)n	6-FAM-GGACCTCGTGAATTACAATCACT	ATCCATTTACCTACCTGTTCATCC
	D16S310	16	(ATAG)n	VIC-GGGCAACAAGGAGAGACTCT	AAAAAAGGACCTGCCTTTATCC
	THO1	11	(TCAT)n	NED-AGGCTCTAGCAGCAGCTCAT	TGTACACAGGGCTTCCGAGT
	IFN-A	9	(GT)n	PET-TGCGCGTTAAGTTAATTGGTT	GTAAGGTGGAAACCCCCACT
MP3	D21S1245	21	(AAAG)n	VIC-CCAGAAAATGACACATGAAGGA	TTGTTGAGGATTTTTGCATCA
	D20S48	20	(GT)n	NED-ATGGTCTCCAGTCCCATCTG	TTGACCTGGATGAGCATGTG
	D9S171	9	(CA)n	NED-TCTGTCTGCTGCCTCCTACA	GATCCTATTTTTCTTGGGGCTA
	D16S476	16	(AAAG)n	6-FAM-GGCAACAAGAGCAAAACTCC	GGTGCTCTCTGCCCTATCTG

PCR Primer pairs for MSA assay. Three multiplexes (MP1, MP2, MP3) were designed to amplify fifteen targets. The 5′ primers are differentially labeled to enable detection by different fluorophores.

**Table 4 ijms-24-13651-t004:** List of triplet (three-tube) multiplex PCR MSI markers for repeats and color channels.

Locus: D4S243	Repeat Type: (ATAG)n	Size Range: Color 165–192 bp	Channel: Blue	K562 -Allele Sizes: 169 bp
FGA	(TTTC)n	299–361 bp	Green	328 bp
D9S747	(GATA)n	179–201 bp	Green	185 bp
Dl 7S654	(CA)n	194–218 bp	Yellow	216 bp
D9S162	(CA)n	117–148 bp	Yellow	143 bp
D17S695	(AAAG)n	170–220 bp	Red	185, 200 bp
MBP & A	(ATGG)n	200–242/119–151	Blue	207, 215, 119 bp
D21S1245	(AAAG)n	209–293 bp	Green	236, 255 bp
D16S310	(ATAG)n	127–170 bp	Green	155, 160 bp
D20S48	(GT)n	251–269 bp	Yellow	261 bp
THOl	(TCAT)n	174–209 bp	Yellow	198 bp
D9Sl 71	(CA)n	109–129 bp	Yellow	126 bp
D16S476	(AAAG)n	176–230 bp	Red	187 209 bp
IFN-A	(GT)n	132–152 bp	Red	no product

**Table 5 ijms-24-13651-t005:** List of triplet (three-tube) multiplex PCR MSI markers and their product size range.

Marker Lower	Limit	Upper Limit
D4S243	0.68	1.33
FGA	0.60	1.43
D9S747	0.63	1.42
Dl 7S654	0.71	1.36
DI 7S695	0.49	1.61
MBP	0.63	1.45
MBPA	0.71	1.37
D16S310	0.6	1.34
D9S162	0.51	1.53
THOl	0.46	1.53
IFN-A	0.68	1.43
D21Sl245	0.58	1.42
D20S48	0.62	1.46
D9Sl 71	0.72	1.36
D16S476	0.54	1.65

## Data Availability

The datasets generated and/or analyzed during the current study are not publicly available as there are no public repositories for this type of dataset. The data are available from the corresponding author on reasonable request.

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
