# Peer review of "Development of Multiplex Polymerase Chain Reaction (PCR)-Based MSA Assay for Bladder Cancer Detection"

_ijms, 2023, doi:10.3390/ijms241713651_

Round 1

Reviewer 1 Report

The title should use the term  "multiplex polymerase chain reaction (PCR) based MSA assay" instead of 3 Multi-PCR Based Microsatellite Instability 2 Analysis (MSA)
English has to be revised in terms of grammar. Extensive language editing is required. Numerous tipplers as well. The constant switching between tenses and their misusage.
Just as an example is the sentence: ". Probably the most deleterious outcome of MSI, either inherited or sporadic sold tumors, is the accumulation of frameshift mutations in tumor-associated genes, which are crucial pathways in human carcinogenesis. 

Normal individuals should be changed to healthy individuals.
Microsatellite instability analysis (MSA) ISN NOT the method for LOH analysis the method is either PCR-polyacrilamide electrophoresis or gen-scan analysis using sequencer. MSA is specific type of analysis that also uses these two methods but for different purposes. MSA and LOH are analyses and gene scan is the method.
Microsatellite instability and LOH are two different genomic instability types, one is microsatellite and other is chromosomal and their etiology is different. Please see the difference in colorectal tumors. Tumors with MIN have completely different path and outcome compared to CIN tumors. The authors have to clearly define in the introduction what is LOH what is the analysis using microsatellite markers. Indeed they can be used in gene scan but this IS NOT microsatellite INSTABILITY analysis. Therefore the name of the whole method that they are describing has to be changed to better reflect that this is not the instability but LOH analysis

(manuscript in prep)- manuscript in preparation cannot be cited
Table 3 color channel depends upon the florescent dye used and it can be changed, therefore this information is neither useful nor relevant.

Extensive editing required.

Author Response

  1. The title should use the term "multiplex polymerase chain reaction (PCR) based MSA assay" instead of 3 Multi-PCR Based Microsatellite Instability 2 Analysis (MSA):      This is changed accordingly as indicated by yellow shadow in revised manuscript

  1. English has to be revised in terms of grammar. Extensive language editing is required. Numerous tipplers as well. The constant switching between tenses and their misusage.Just as an example is the sentence: ". Probably the most deleterious outcome of MSI, either inherited or sporadic sold tumors, is the accumulation of frameshift mutations in tumor-associated genes, which are crucial pathways in human carcinogenesis.       Extensive language editing including above mentioned paragraph was performed in revised manuscript as indicated by yellow shadow throughout entire revised manuscript.

  1. Normal individuals should be changed to healthy individuals.:    This is changed accordingly as indicated by yellow shadow throughout entire manuscript.

  1. Microsatellite instability analysis (MSA) ISN NOT the method for LOH analysis the method is either PCR-polyacrilamide electrophoresis or gen-scan analysis using sequencer. MSA is specific type of analysis that also uses these two methods but for different purposes. MSA and LOH are analyses and gene scan is the method. Microsatellite instability and LOH are two different genomic instability types, one is microsatellite and other is chromosomal and their etiology is different. Please see the difference in colorectal tumors. Tumors with MIN have completely different path and outcome compared to CIN tumors. The authors have to clearly define in the introduction what is LOH what is the analysis using microsatellite markers. Indeed they can be used in gene scan but this IS NOT microsatellite INSTABILITY analysis. Therefore the name of the whole method that they are describing has to be changed to better reflect that this is not the instability but LOH analysis.        These issues were properly addressed in line line 28 to 44 in page 1 and line 70-75, line 80 in page 2 as indicated by yellow shadow high light.
  2. (manuscript in prep)- manuscript in preparation cannot be cited: This is properly removed accordingly

  1. Table 3 color channel depends upon the florescent dye used and it can be changed, therefore this information is neither useful nor relevant.: This table is taken out from the revised manuscript.           This is properly removed accordingly, and all other tables are accordingly re numbered properly as outlined with yellow color highligt

Of note reviewer 2 suggested revisions are marked as green color shadow.

Reviewer 2 Report

The paper seems relevant and interesting and is reasonably well written except for the grammar.  The results, discussion and methodology of the doublet reaction should be deleted as it is not relevant to the paper, since this is an assay which could not be validated.  It seems to be filler.

The triplet assay should be validated on more than 6 positive cases.  The typical lab validation would be 20-30 specimens of which at least half would be positives.  Not sure why more cases are described in methods than used for the validation?  As well, I would like to see a figure of an electrophoragram as to what the assay actually looks like on the analyzer.  There will be space for this figure when the doublet stuff is deleted.

I would prefer that the overall results table not use numbers to describe the results, but use text for the descriptions (eg N, negative; P; positive; NI, non-informative; U, uninterpretable/non-evaluable) as it is currently a bit confusing and non-intuitive.

Also, there are numerous significant and minor (definite and indefinite articles) grammatical errors in all sections of the manuscript, so the MS needs better editing.  I have identified some examples in the appended marked up pdf.

Lots of English grammatical errors.  Better editing needed.

Author Response

The paper seems relevant and interesting and is reasonably well written except for the grammar.  The results, discussion and methodology of the doublet reaction should be deleted as it is not relevant to the paper, since this is an assay which could not be validated.  It seems to be filler.

  • These have been deleted and main texts are appropriately modified outlined as yellow shadow highlight

The triplet assay should be validated on more than 6 positive cases.  The typical lab validation would be 20-30 specimens of which at least half would be positives.  Not sure why more cases are described in methods than used for the validation?  As well, I would like to see a figure of an electrophoragram as to what the assay actually looks like on the analyzer.  There will be space for this figure when the doublet stuff is deleted. This has been mentioned in abstract result and discussion

  • In Page 6, From this final triplet assay, we have then repeated multiple times on various samples in-cluding original 25 cancer sample and 15 healthy control samples (JHU samples) and re-producibility has been consistent. Finally, we have performed, in non-blind fashion, to test 20 QC samples from EDRN study by our newly established assay (blue color).
  • In Page 8, increased number of cancer samples (We have used 25 JHU samples and 6 EDRN sam-ples), we could have enhanced (green color highlight)

I would prefer that the overall results table not use numbers to describe the results, but use text for the descriptions (eg N, negative; P; positive; NI, non-informative; U, uninterpretable/non-evaluable) as it is currently a bit confusing and non-intuitive.

  • We agree with this point. However as the table was made as a figure and converted to PDF file it was very difficult to change it. We apologize for this in situation

Also, there are numerous significant and minor (definite and indefinite articles) grammatical errors in all sections of the manuscript, so the MS needs better editing.  I have identified some examples in the appended marked up pdf.

  • This has been widely edited from first and second reviewer comments. The comments from reviewer 3 was all edited and highlighted with yellow color

Reviewer 3 Report

The article by Reynolds et al., "Development of 3 Multi-PCR-Based Microsatellite Instability Analysis (MSA) for Bladder Cancer Detection", presents the data for the development of a triplet MSA assay using 15 markers with a genetic analyzer. Their Triple MSA correctly predicted all of 6/6 cancer samples to be cancerous samples and 14/14 samples from normal individuals to be normal samples. These findings are promising, and it could be a potentially time- and cost-effective genetic assay for bladder cancer detection.

Below, I provide comments to improve this article.

  1. It would be beneficial to add more cancer samples for this study. The sample size is too small.
  2. Table 2+3: Please mention the name of the gene related to each locus.
  3. The authors should provide clinicopathological information on the patient sample used for this study. The information should include age, sex, smoking history, clinical stage of the disease, whether the tumors are muscle invasive or non-invasive, etc.
  1. Is it possible to provide the MSI status of the cancer samples?
  2. Please provide the disease progression history of the patients who tested positive for cancer by Triple MSA.
  3. The authors should perform IHC to check the expression of MMR proteins (hMLH1, hPMS2, hMSH2, and hMSH6) to further validate this data set.

Minor editing of English language required.

Author Response

It would be beneficial to add more cancer samples for this study. The sample size is too small.

  • We were provided with only limited amount of samples for both JHU samples and EDRN samples. We have discussed about this shortcomings in page 10. With green colored highlight

Table 2+3: Please mention the name of the gene related to each locus.

  • Table 3 was removed as a result of reviewer 1 suggestion. In Table 2 title section name of available gene was outlined

The authors should provide clinicopathological information on the patient sample used for this study. The information should include age, sex, smoking history, clinical stage of the disease, whether the tumors are muscle invasive or non-invasive, etc.

  • We were NOT provided with these important informations for both JHU samples and EDRN samples. We have discussed about this shortcoming in page 10 and briefly in abstract. with green colored highlight

Is it possible to provide the MSI status of the cancer samples?

- We were NOT provided with proper samples for both JHU samples and EDRN samples for analysis. We have discussed about this shortcoming in page 10. with green colored highlight

It would be beneficial to add more cancer samples for this study. The sample size is too small.

  • We were provided with only limited amount of samples for both JHU samples and EDRN samples. We have discussed about this shortcoming in page 10. With green colored highlight

Table 2+3: Please mention the name of the gene related to each locus.

  • Table 3 was removed as a result of reviewer 1 suggestion. In Table 2 title section name of available gene was outlined

The authors should provide clinicopathological information on the patient sample used for this study. The information should include age, sex, smoking history, clinical stage of the disease, whether the tumors are muscle invasive or non-invasive, etc.

  • We were NOT provided with these important informations for both JHU samples and EDRN samples. We have discussed about this shortcoming in page 10 and briefly in abstract. with green colored highligh

Is it possible to provide the MSI status of the cancer samples?

- We were NOT provided with proper samples for both JHU samples and EDRN samples for analysis. We have discussed about this shortcoming in page 10. with green colored highlight             

Please provide the disease progression history of the patients who tested positive for cancer by Triple MSA.

  • We were NOT provided with these important informations for both JHU samples and EDRN samples. We have discussed about this shortcoming in page 10 and briefly in abstract. with green colored highlight

The authors should perform IHC to check the expression of MMR proteins (hMLH1, hPMS2, hMSH2, and hMSH6) to further validate this data set.

  • We were NOT provided with proper samples for both JHU samples and EDRN samples for this important analysis. We have discussed about this shortcoming in page 10. with green colored highlight

The authors should perform IHC to check the expression of MMR proteins (hMLH1, hPMS2, hMSH2, and hMSH6) to further validate this data set.

  • We were NOT provided with proper samples for both JHU samples and EDRN samples for this important analysis. We have discussed about this shortcoming in page 10. with green colored highlight

Comments on the Quality of English Language

Minor editing of English language required.

  • English is extensively edited as highlighted with yellow color in revised paper

Round 2

Reviewer 1 Report

no further comments

no further comments

Author Response

Overall Language is edited again and highlited with yellow and green color at attached files.

Reviewer 2 Report

1.     The manuscript is much improved from the first version.  I am still confused, though, by the number of specimens tested.  Non-blinded specimens used for assay development do not traditionally count as a validation specimen.  Since this manuscript has been written to propose a new method of testing patient specimens for bladder cancer, replacing other methods, it is important to identify the quality of the assay.  From my interpretation of the manuscript, it still seems as though only 6 positive patient specimens have been tested in a blinded fashion (lines 18-25).  If more patient specimens have been tested in a blinded manner, reproducing actual testing (more or less) then this needs to be better explained in the manuscript--certainly in the abstract.  If only 6 positive specimens have been utilized of the 20 blind tests, I would personally like to see more positives tested (say half of the 20, or even better 12-15 of 30).  Robustness and reliability of clinical assays is important.

2.     The discussion of lines 263-277 is quite confusing.  It needs to be framed in a way that is relevant to the aim of the manuscript, and written to be more easily understood.

3.     While I don't quite understand the difficulty in changing the values of the spreadsheet of Figure 1 and then reprinting the pdf (which might only take a few minutes), I can live without the changes being made (though I still think that the Figure would be improved with the changes).

Author Response

  1. The manuscript is much improved from the first version. I am still confused, though, by the number of specimens tested.  Non-blinded specimens used for assay development do not traditionally count as a validation specimen.  Since this manuscript has been written to propose a new method of testing patient specimens for bladder cancer, replacing other methods, it is important to identify the quality of the assay.  From my interpretation of the manuscript, it still seems as though only 6 positive patient specimens have been tested in a blinded fashion (lines 18-25).  If more patient specimens have been tested in a blinded manner, reproducing actual testing (more or less) then this needs to be better explained in the manuscript--certainly in the abstract.  If only 6 positive specimens have been utilized of the 20 blind tests, I would personally like to see more positives tested (say half of the 20, or even better 12-15 of 30).  Robustness and reliability of clinical assays is important.

  • Only 6 cancer samples out of 20 total samples were given and tested from EDRN samples. 

    Of note, this prospective multi-center study of a new diagnostic test for microsatellite analysis (MSA) for bladder cancer will include 12 sites throughout the United States and Canada.

    Three groups were included in this study.  Two of the groups include 200 participants without bladder cancer and served as control groups.  These two control groups will be broken down into two cohorts.  The first cohort, Control Group 1, included 50 male and 50 female participants without gastroenterology/urology (GU) disease with normal urinalysis and cytology determinations.  The second cohort, Control Group 2, included 100 participants with one of four disease processes [benign prostatic hypertrophy (BPH), foreign body (urinary stones, stents, and catheters), infection, and hematuria], which have historically led to false positive urinary bladder cancer screening studies.

    The third group included 300 participants with incident or recurrent bladder tumors who are followed every 3 months with cytology and cystoscopic examinations for recurrence of their bladder tumors.  MSA results were compared with these standard examinations for recurrent disease.  The results of MSA were evaluated for both recurrent tumors as well as for anticipation of subsequent recurrent disease.

    The investigators were blinded to the MSA assay results.  No clinical decision regarding medical care or management were based upon the MSA assay results alone. 

    The contract laboratory responsible for performing the MSA assay was blinded to all participant urinalysis, cytology, cystoscopic and pathology results to minimize bias. The contract laboratory performed a qualification blinded test before the assay was put into use for the Validation study.

    The report we present is coming from the contract laboratory responsible for performing the MSA assay and includes the results from the MSA assay using Qualification blinded samples to assure the accuracy of the assay before its use in the clinical trial Validation study discussed above. We wish to publish these results as part of our efforts to report findings from the EDRN Validation clinical trial in a follow up paper which includes a statistical analysis of each marker including sensitivity, specificity and accuracy analysis and a refining of the assay to include only the most informative markers.

  • We clarified this page 8 discussion 246 to 253 line outlined with ponk color

  1. The discussion of lines 263-277 is quite confusing. It needs to be framed in a way that is relevant to the aim of the manuscript, and written to be more easily understood.
  • Lines 263-277 are not discussion section. Can u clarify this ? 
  • We were assuming that you were referring to the newly edited part line 241 to 244 highlighted with green color page 8 discussion
